# MixLLM: Selecting Large Language Models with High-Quality Results and Minimum Inference Cost for Multi-Stage Complex Tasks

## Abstract

We study the problem of selecting LLMs for multi-stage complex tasks to jointly optimize final result quality and minimize LLM inference cost. Existing approaches primarily target simple tasks or optimize for improving result quality or reducing cost only, overlooking the trade-off between them. We address this gap by systematically investigating LLM performance in multi-stage workflows. Inspired by our findings from real-world applications, we formalize the LLM selection problem as a constraint-based optimization task with good properties: guarantee lower bounds on accuracy, minimize LLM inference cost and tolerate performance fluctuations caused by LLM stochasticity, making it more practical for users. We further introduce MixLLM, a search framework that leverages the exploration–exploitation principle to adaptively balance result quality and LLM inference cost. MixLLM is carefully designed to efficiently identify a (near-)optimal solution with minimal exploration and to terminate automatically and early via search-space pruning. Experimental results demonstrate that, compared to using a single powerful commercial or open-source LLM, or selecting LLMs via existing state-of-the-art methods, our approach not only improves result quality (by $1\% - 16\%$) but also significantly reduces inference cost (by $18\% - 92\%$). In addition, our approach efficiently adapts to different tasks, methods, and datasets, demonstrating its practicality and robustness for multi-stage complex tasks.

## 1 Introduction

Large language models (LLMs), with excellent context understanding, generalization and reasoning capabilities, revolute how users interact with data, process information, and derive valuable insights, driving advancements in applications such as text-to-SQL (Pourreza & Rafiei, 2023), data cleaning (Qian et al., 2024), and sentiment analysis (Patel et al., 2024). To ensure good performance and stability, developers commonly decompose complex tasks into multi-stage workflows, utilizing LLMs or tools at each stage to generate intermediate results that contribute to final outputs.

With the prevalence of multi-stage tasks, selecting the appropriate LLM for each stage emerges as an important problem. The most straightforward approach is to apply a single fixed LLM (*e.g.*, the latest OpenAI model) across all stages, however, it misses the opportunity to leverage different LLMs to improve quality, reduce cost, or achieve a balance. The pricing of LLMs varies significantly across vendors, and expensive models do not always outperform cheaper ones (Mod, 2025). This heterogeneity in cost and accuracy presents a significant challenge, as well as an opportunity, in selecting the optimal LLM for each stage to jointly optimize quality and cost.

Existing LLM selection methods (*cf.* Table 1) such as binary routers (Ong et al., 2024; Ding et al., 2024), cascade pipelines (Yue et al., 2024; Chen et al., 2024), and multi-choice routers (Feng et al., 2025) mainly targets simple question-answering tasks, not complex multi-stage workflows. Although Chen et al. (2025) proposes LLMSelector for multi-stage tasks, it focuses solely on accuracy optimization, neglecting the inference cost. In summary, existing work does not address the quality-cost trade-off in LLM selection for multi-stage tasks.

**Challenges.** Selecting LLMs per stage introduces fundamental challenges: **(C1)** In multi-stage tasks, the result quality depends on the final outputs, and the inference cost depends on the effi-

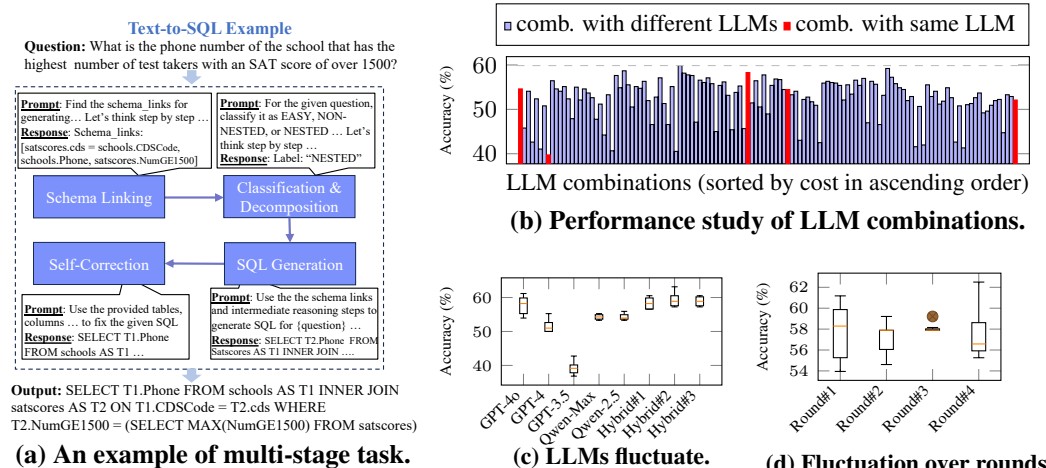

(a) An example of multi-stage task.

(b) Performance study of LLM combinations.

(c) LLMs fluctuate.

(d) Fluctuation over rounds.

Figure 1: **(a)** an example of multi-stage text-to-SQL (Pourreza & Rafiei, 2023). **(b)** each bar represents an LLM combination, ordered by cost. *It is better to apply different LLMs rather than the same LLM in order to improve result quality and/or reduce inference cost.* **(c)** the result accuracy of LLMs fluctuates. **(d)** accuracy is reported over four rounds, each using the same LLM to process queries five times. The fluctuation ranges of different rounds overlap significantly and the mean values are close, indicating that *the accuracy of LLMs fluctuates within a certain range and this variability is relatively stable.* **(b)(c)(d) present some findings, see more in Appendix A.**

Table 1: Comparison of MixLLM with existing LLM routers or selectors (details in Section 2).

| LLM Selection Methods | Support for Multiple LLMs | Designed for Multi-Stage Tasks | Co-Optimization of Accuracy and Cost | Free of Model Training |
|---|---|---|---|---|
| HybridLLM | ✗ | ✗ | ✓ | ✗ |
| RouteLLM | ✗ | ✗ | ✓ | ✗ |
| FrugalGPT | ✓ | ✗ | ✓ | ✗ |
| LLMCascade | ✓ | ✗ | ✓ | ✓ |
| GraphRouter | ✓ | ✗ | ✓ | ✗ |
| LLMSelector | ✓ | ✓ | ✗ | ✓ |
| **MixLLM (Ours)** | ✓ | ✓ | ✓ | ✓ |

ciency of each stage. Therefore, monotonically optimizing both the accuracy and inference cost of a single stage is neither a good idea nor an appropriate approach. **(C2)** Furthermore, there is no clear relationship between the accuracy and the cost of different LLM combinations (*cf.* Figure 1(b), see details in Appendix A). This poses challenges in measuring the goodness of each LLM combination to decide the holistic optimization direction. **(C3)** Even worse, the interplay between LLMs across stages, combined with their inherent complexity and opacity (Vaswani et al., 2017), renders performance modeling exceptionally challenging. **(C4)** The performance of LLM combinations varies across different tasks, methods, and datasets (*cf.* see details in our evaluation results in Section 4), highlighting the need for a more generalizable solution.

In this paper, we address the problem of selecting LLMs for multi-stage tasks to jointly optimize final result quality and minimize inference cost. Our approach is motivated by the observation that strategically combining different LLMs can achieve high-quality results while significantly reducing cost (*cf.* Figure 1(b)). We next introduce our contributions.

**Our Contributions.** We first investigate performance characteristics of LLMs in multi-stage tasks. Based on observations, we formulate LLM selection problem from a practical perspective and propose methods to select appropriate LLMs for each stage, optimizing accuracy and inference cost.

• We formalize the LLM selection problem as a constraint-based single-objective optimization problem with good properties: guaranteeing lower bounds on accuracy, minimizing inference cost and tolerating performance fluctuations. LLMs theoretically exhibit performance fluctuations due to inherent stochasticity (Vaswani et al., 2017). Our performance study empirically validates this prop-

erty, while further indicating that such variability is relatively stable (*cf.* Figure 1(b)(c)). Inspired by this property, we adopt the performance fluctuation range as a relaxation factor for a baseline level of accuracy, which extends the spectrum of LLM combinations with comparable accuracy and lower inference cost.

• We propose MixLLM, a tree-based search method that treats LLM performance as a black box and adaptively refines itself using exploration and exploitation, to address the aforementioned challenges. To further tackle the challenge *C2*, MixLLM utilizes a simple but effective two-phase greedy strategy: once it finds a node with high-quality results, it further explores the possibility of reducing cost, adaptively balancing accuracy and cost. MixLLM efficiently identifies a (near-)optimal solution with minimal exploration, terminates automatically and early through space pruning.

• We extensively evaluate our approach on three tasks: text-to-SQL, data imputation and fact checking, covering eight different multi-stage workflows and four popular datasets. Compared to state-of-the-art methods (*e.g.* FrugalGPT, GraphRouter, LLMSelector), our method can improve result accuracy by 1% – 16% while simultaneously reducing inference cost by 18% – 92%. Our experiments also demonstrate its practicality and robustness across diverse tasks, methods, and datasets.

## 2 RELATED WORK

**LLM Routers or Selectors.** Binary routers such as HybridLLM (Ding et al., 2024) and RouteLLM (Ong et al., 2024) train routers based on task-specific labeled data to select between a stronger LLM and a weaker LLM. However, they cannot support model selection across many LLMs. LLM cascading methods such as FrugalGPT (Chen et al., 2024) and LLMCascade (Yue et al., 2024) organize all candidate LLMs in a list by their cost and/or accuracy and then assigns a query to the list of LLMs sequentially if the answers generated by the prior LLMs are unacceptable. FrugalGPT trains a score function, while LLMCascade relies on the answer consistency of the weak LLMs across multiple samples. GraphRouter (Feng et al., 2025) trains a router among multiple LLMs using a graph neural network. While these methods address the accuracy-cost trade-off , they are primarily designed for simple question-answering tasks by selecting an LLM for each question. In contrast, our approach targets complex tasks with multi-stage workflows and aims to assign potentially different LLMs to each stage. Similar to our scenario, LLMSelector (Chen et al., 2025) selects LLM for each stage to improve the accuracy of final results based on two monotonic assumptions, but ignores inference cost. Unlike LLMSelector, we address the challenge of optimizing both accuracy and inference cost without making any assumption. Table 1 summarizes the described differences between our method and the aforementioned methods.

**Other Related Methods.** Bayesian optimization (*e.g.* Cog (2024)) takes the optional LLMs at each stage as parameters and identify pareto-optimal solutions. However, users still face the challenge of determining an appropriate number of iterations for termination and selecting a final solution from a multitude of Pareto-optimal candidates. Unlike Bayesian optimization, our approach terminates automatically and outputs the final solution without further involvement of user. UCT (Upper Confidence Bounds applied to Trees) (Kocsis & Szepesvári, 2006) provides a general tree search framework that leverages exploration and exploitation. However, it does not address the joint optimization of accuracy and inference cost for multi-stage tasks, and it also relies on a fixed number of iterations for termination. In contrast, our method overcomes these limitations.

## 3 MIXLLM: LLM SELECTION FOR MULTI-STAGE COMPLEX TASKS

We systematically investigate the actual performance characteristics of LLMs in the multi-stage complex tasks. Due to space limitations, we put detailed findings in Appendix A. In this section, we focus on our problem formulation and method, which are inspired by our findings.

### 3.1 PROBLEM FORMULATION

Defining an optimization objective that jointly considers result quality and inference cost is challenging, given their unclear relationship (see challenge *C2* in Section 1). However, our empirical

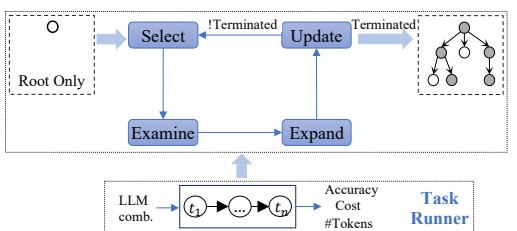

Figure 2: MixLLM Framework

1   $v^* \leftarrow \emptyset$
2   **while** *!Terminate()* **do**
3      $v \leftarrow$ Select()
4      $A(v), C(v) \leftarrow$ Examine()
5      **if** $A(v) \geq a - \theta$ *and* $C(v) < C(v^*)$ **then**
6        $v^* \leftarrow v$
7      Expand($v$)
8      Update()
9   **return** $v^*$

analysis reveals a critical insight: while performance fluctuation inherently exists in LLMs due to stochasticity (Vaswani et al., 2017), this variability is relatively stable (*cf.* Figure 1(c)(d)). This motivates us to use the performance fluctuation range $\theta$ as a relaxation factor. Furthermore, we observed that the best LLM combination using the same LLM throughout all stages can consistently offer high-quality results at a potentially high cost (*cf.* Figure 1(b), details in Appendix A), it could serve as a baseline level of accuracy $a$. By relaxing the baseline level of accuracy $a$ with fluctuation range $\theta$, the problem covers a number of LLM combinations with comparable accuracy but lower inference cost (*cf.* Figure 1(b)). Based on these, we next describe our problem definition.

**Definition 1** (LLM Selection Problem). *Given a task $T = (t_1, t_2, \ldots, t_k)$ consisting of $k$ stages and a set of candidate LLMs $\mathcal{L}$, the problem is to identify an LLM combination $M = (m_1, m_2, \ldots, m_k)$ that satisfies the following equation:*

$$
\begin{aligned}
\min \quad & C(M) \\
s.t. \quad & A(M) \geq a - \theta.
\end{aligned}
\tag{1}
$$

*$C(M)$ and $A(M)$ denote the cost and accuracy of applying $M$ to solve task $T$, respectively. The parameter $a$ specifies the accuracy of a baseline high-quality solution, e.g., $a = \max_{c \in C_{same}} A(c)$ ($C_{same}$ is the set of LLM combinations using the same LLM across all stages, and $A(c)$ is the accuracy of a combination $c$). The parameter $\theta$ refers to a reasonable fluctuation range to flexibly control the result quality, e.g., $\theta = \mathrm{Std}\left(\left\{A(c)_{(i)}\right\}_{i=1}^{n}\right)$, where $i$ denotes a trial. Besides our default setting, both $a$ and $\theta$ could be determined empirically, either through user experience on applications or evaluation on sampled data.*

**Parameter $\theta$ and Benefits.** We discuss three occasions:

1. $\theta > 0$: this is the default setting described above. The accuracy constraint is relaxed from baseline $a$ to $a - \theta$, allowing more combinations to be considered, potentially reducing inference cost with acceptable accuracy. Here $\theta$ is a small value (*e.g.* standard deviation) designed to tolerate the performance fluctuation of LLMs and ensure that the accuracy within the range $[a - \theta, a]$ is still reasonably good for the task.

2. $\theta = 0$: this means that we only consider LLM combinations whose accuracy is strictly not less than an empirical value $a$. This often occurs when users have higher requirements on the result quality, so we preserve the accuracy to be no less than the default solution.

3. $\theta < 0$: this indicates a higher requirement for the performance of LLM combinations, whose accuracy must exceed $a + |\theta|$. It happens when a sufficient number of LLM combinations obtain higher accuracy than the default in the task. At this time, we can filter some LLM combinations, but there is also an opportunity to discover some LLM combinations that perform better in terms of both accuracy and cost (*e.g.*, Finding 4 in Appendix A).

### 3.2 MixLLM Framework and Techniques

**Search Tree.** We build a tree to represent possible LLM combinations, where each node $v$ corresponds to an LLM combination [1]. The tree starts from a single root node (*e.g.* the combination of the same LLM with the highest accuracy) and it expands by applying possible actions (*cf.* Definition 2) to dynamically generate child nodes.

**Definition 2** (Action $Act(v, j, \ell)$). *Given a set of candidate LLMs $\mathcal{L}$ for a $k$-stage task $T$, a node $v$ and corresponding LLM combination $v = (v_1, v_2, \ldots, v_k)$ ($v_i \in \mathcal{L}$), and an LLM $\ell$ ($\ell \in \mathcal{L}$), we*

---

[1]With no ambiguity, in the paper, node and LLM combination are used interchangeably, both denoted as $v$.

*define an action $Act(v, j, \ell) = (v_1, \ldots, v_{j-1}, \ell, v_{j+1}, \ldots, v_k)$, which is the process to derive a child node $v' = (v_1, \ldots, v_{j-1}, \ell, v_{j+1}, \ldots, v_k)$ by changing the $j$-th stage of $v$ from $v_j$ to $\ell$, where $v_j \neq \ell$.*

Recall that we use $A(v)$ and $C(v)$ to denote the exact result accuracy and the inference cost of invoking the test task with the LLM combination on $v$, respectively. Our goal is to find the optimal node $v^*$ whose $A(v^*)$ is not less than $a - \theta$ and $C(v^*)$ is minimized (*cf.* Equation (1)).

**MixLLM Overview.** Figure 2 illustrates the overall framework, which includes two components: *task runner* and *MixLLM engine*. The task runner works as a service to examine LLM performance. It takes an LLM combination as input, executes it on the given task, and outputs the final accuracy, the number of processed tokens and the inference cost. MixLLM engine searches for the optimal solution based on the following algorithm framework (*cf.* Algorithm 1): (1) **Select (line 3)**: selects a *promising* node $v$ that has high possibility to satisfy accuracy constraint and reduce inference cost, see details below. (2) **Examine (line 4)**: examines the exact accuracy $A(v)$ and cost $C(v)$ of a node $v$ by invoking its corresponding combination of LLMs to complete the task. Examined values are used to update the optimal solution $v^*$ found so far (line 5–6). (3) **Expand (line 7)**: expands the current node $v$ with possible promising nodes by applying actions (*cf.* Algorithm 2 in Appendix B). (4) **Update (line 8)**: identifies unpromising nodes to prune space based on exact accuracy and cost of examined nodes; (5) **Terminate (line 2 and 9)**: terminates *automatically* if there is no promising node, and finally returns the optimal solution found so far. More algorithmic details will be presented below. Before that, we introduce the principles used to enrich our algorithm.

We want to find an LLM combination that preserves high-quality results with near-minimum inference cost while reducing exploration overhead. To address the aforementioned challenges introduced in Section 1, we propose two fundamental **principles** to enrich our algorithm:

1. *Examining as few LLM combinations as possible to discover the (near-)optimal combination.*

2. *When the (near-)optimal combination has been discovered, the algorithm should automatically terminate as early as possible.*

The first principle aims to select potential combinations in an earlier stage, but does not indicate the termination of exploration, as we cannot ensure the optimality of a combination without complete solution space. The principle requires a strategy that collaborates both accuracy and cost to guide the optimization direction. The second principle complements the first principle by emphasizing the importance of early termination. It demands techniques to identify unpromising combinations, which are critical for combinatorial space pruning and automatic termination, as persistent exploration inevitably incurs prohibitive overhead. Only by combining these two principles can we find a (near-)optimal solution and simultaneously reduce search overhead.

**Select (for Principle 1): navigating search direction via a two-phase greedy strategy.** The *select* step is critical for achieving the first principle, which requires measuring the goodness of LLM combinations based on both result accuracy and inference cost. However, due to challenges (*C1* to *C4*) in Section 1, jointly modeling accuracy and cost via a reward function is impractical. To address this, we propose a simple but effective two-phase greedy strategy. Specifically, if MixLLM identifies a node with satisfactory accuracy, then it focuses on minimizing inference cost; otherwise, it selects the node expected to deliver the greatest cost reduction. Each node utility comprises two components: cost utility, based on model pricing and profile data such as token count; and accuracy utility, based on the well-known upper confidence bound applied to trees (UCT, Kocsis & Szepesvári (2006)). This design enables MixLLM to adaptively balance accuracy and inference cost in node selection. Notably, by prioritizing cost minimization once the accuracy constraint is satisfied, MixLLM facilitates space pruning, thus achieving the second principle.

**Terminate (for Principle 2): automatic and early termination through node pruning.** The second principle emphasizes the importance of automatic and early termination. Note that a node $v$ is *unpromising* and can be pruned if its cost $C(v)$ exceeds that of the best node found so far and/or its accuracy $A(v)$ does not meet the accuracy constraint. To avoid the prohibitive cost of *examining* nodes, MixLLM estimates cost and accuracy, then use them for combinatorial space pruning.

- *Accuracy Prediction:* Predicting the accuracy of a node is difficult due to the challenges (*C1*) in Section 1. Inspired by the lineage of nodes in search tree, MixLLM uses runtime experience knowledge to predict the trend of relative accuracy, a binary value indicating whether an unexplored node $v$ has the potential to surpass the accuracy of its examined parent node $u$. To

further improve prediction reliability in the presence of LLM stochasticity, MixLLM aggregates the predicted binary values between node $v$ and its multiple neighbors, producing the final prediction value $p(v)$ that indicates whether node $v$ could potentially outperform *some* of its evaluated neighbors.

- *Cost Estimation:* We utilize profiling data (*e.g.*, number of processed tokens) to estimate inference cost. For example, by profiling combinations that use the same LLM, the required profiling space is significantly reduced compared to the full combinatorial space. Our experiments demonstrate that such a simple strategy can achieve a relatively high level of accuracy (*cf.* Section 4.6).

We consider a node to be unpromising if the predicted accuracy is much lower or the estimated cost is much higher than the best combination explored so far. MixLLM delays the removal of unpromising nodes by updating node status during exploration, it terminates when there are no promising nodes. This enables finding a (near-)optimal solution while reducing the search space.

## 4 EXPERIMENTAL EVALUATION

### 4.1 EXPERIMENTAL SETUP

**LLM-based Tasks, Methods, and Datasets.** We evaluate three tasks, namely text-to-SQL, data imputation and fact checking, covering eight different multi-stage workflows and four datasets. For text-to-SQL, we employ three different multi-stage methods: MAC_SQL (Wang et al., 2024), DIN_SQL (Pourreza & Rafiei, 2023) and E_SQL (Caferoğlu & Özgür Ulusoy, 2025), and conduct experiments on both Spider (Yu et al., 2018) and BIRD (Li et al., 2023) — two widely-used benchmarks where BIRD is much more complex (Li et al., 2024). The above three text-to-SQL methods cover not only the differences in internal details of each stage, but also the differences in external structures across all stages. We use these methods and benchmarks to verify the robustness of our approach across different workflows and datasets for the same task. For data imputation task, we employ UNIDM (Qian et al., 2024) — a general multi-stage based data cleaning framework, and the popular and challenging Restaurant datasets. For fact checking task, we employ the multi-stage workflow composed of LOTUS (Patel et al., 2024) and the well-known FERVER benchmark dataset (Thorne et al., 2018). Due to space limitation, we refer interested readers to the above relevant literature for details of tasks, workflows or datasets.

**Baselines.** We compare MixLLM with the following baselines. Note that we do not compare with binary routers (*e.g.* HybirdLLM, RouteLLM) because they do not support multiple LLMs.

- **ELLM**: selects the most expensive candidate LLM (a powerful commercial model from OpenAI).
- **CLLM**: selects the cheapest candidate LLM (a powerful open-source model).
- **FrugalGPT** (Chen et al., 2024) : an instantiation of LLM cascade, it trains a score function in advance and then predicts which LLMs to use in order to achieve quality-cost trade-off.
- **GraphRouter** (Feng et al., 2025) : trains a router among multiple LLMs using a graph neural network, in order to achieve quality-cost trade-off.
- **LLMSelector** (Chen et al., 2025): uses a greedy strategy to select LLM for each stage in order to improve the accuracy of final results based on monotone assumption, but ignores inference cost.

We also compare MixLLM with other search-based methods on our formulated problem.

- **BOS**: emploies bayesian optimization to find appropriate LLMs.
- **UCT**: a tree search method based on the upper confidence bounds (Kocsis & Szepesvári, 2006).

**Candidate LLMs.** The set of candidate LLMs includes GPT-4-turbo, GPT-4o, GPT-3.5-turbo, Qwen-2.5, and Qwen-Max, covering state-of-the-art commercial and open-source models. To ensure reproducibility, we set the temperature to zero, run LLMs on all evaluated tasks and repeat five times. Based on these runs, all methods obtain the same average accuracy and cost of LLMs.

### 4.2 END-TO-END COMPARISON WITH BASELINES ACROSS DIFFERENT MULTI-STAGE TASKS.

Figure 3 compares the accuracy of final results and the inference cost of LLMs selected by MixLLM and baselines in three different multi-stage tasks. In the figure, red bars represent the percentage of

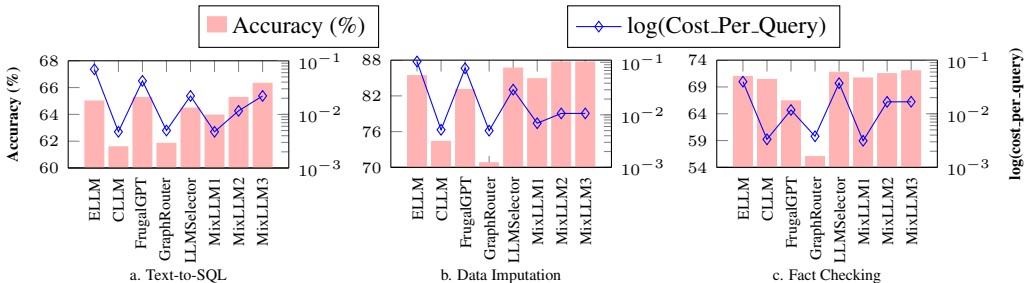

Figure 3: Accuracy-cost tradeoff for MixLLM compared to baselines on different multi-stage tasks.

correct results, and blue marks represent the average per-query cost (in a log scale) of invoking corresponding LLMs to generate results. For our solution MixLLM, we evaluate three cases by varing the values of $\theta$, as discussed in Section 3.1: (1) $\theta$ is the default value (*i.e.* standard deviation); (2) $\theta = 0$; (3) $\theta = -1$, represented by MixLLM1, MixLLM2, and MixLLM3, respectively.

MixLLM achieves comparable or higher accuracy than baselines with much lower LLM inference cost. Specifically, compared to FrugalGPT and GraphRouter, MixLLM can improve result accuracy by $1.1\% – 16.9\%$ and reduce inference cost by $18\% – 91\%$. Against LLMSelector, which is designed to improve result accuracy for multi-stage tasks, MixLLM2 and MixLLM3 achieve a $0.3\% – 1.8\%$ improvement in accuracy with a $47\% – 65\%$ reduction in inference cost; alternatively, MixLLM1 incurs an accuracy loss of $0.5\% – 1.8\%$ but decreases the inference cost by $77\% – 92\%$. In addition, compared to applying the most powerful and expensive commercial LLM from candidates (*i.e.* ELLM), MixLLM achieves comparable or even higher accuracy (up to $2.3\%$ improvement), while significantly reducing the inference cost by up to $93\%$. Compared with the most cost-effective LLM (CLLM), MixLLM1 can offers a significant improvement in accuracy ($2.4\% – 10.5\%$) with no additional or only a modest increase in inference cost ($0\% – 33\%$). Among our approaches, MixLLM1 generally achieves near-highest accuracy at minimal cost, MixLLM2 obtains higher accuracy with higher cost, and MixLLM3 may increase both accuracy and inference cost, validating the benefits of three occasions for $\theta$ as discussed in Section 3.1. These results demonstrate that MixLLM can effectively balance accuracy and LLM inference cost on various multi-stage tasks.

## 4.3 COMPARISON WITH SEARCH-BASED APPROACHES UNDER OUR PROBLEM DEFINITION

Figure 4 compares MixLLM with BOS and UCT on our defined LLM selection problem (*cf.* Definition 1). Typically, we uses red lines to represent the value of accuracy constraint defined by the problem. Results above red lines indicate that the accuracy requirement is satisfied, and among these, lower inference cost corresponds to better performance. MixLLM demonstrates superior cost-accuracy trade-offs compared to evaluated methods. Specifically, it achieves a $0\% – 0.5\%$ improvement in accuracy while simultaneously reducing inference cost by $20\% – 72\%$ compared to BOS, and delivers $19\% – 89\%$ cost reduction compared to UCT, with comparable or higher accuracy (up to $1.5\%$ improvements).

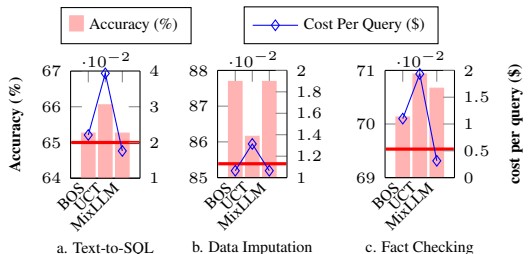

Figure 4: Comparing MixLLM with search-based approaches under our formulated problem.

## 4.4 ADAPTING TO VARIOUS WORKFLOWS AND DATASETS UNDER THE SAME TASK

Figure 5 presents the result accuracy and inference cost of LLMs found by MixLLM (the three variants used in Section 4.2) and baselines, across three multi-stage workflows and two datasets for the text-to-SQL task. MixLLM achieves superior cost-accuracy trade-offs over baselines. Specifically, in terms of result accuracy, MixLLM variants outperform FrugalGPT by $1.1\% – 8.6\%$, GraphRouter by $0.9\% – 21.8\%$, LLMSelector by $1.2\% – 9.8\%$, ELLM by $1.3\% – 15.3\%$ and CLLM by $1.2\% – 12.9\%$. As for inference cost, MixLLM variants reduce cost by $74\% – 89\%$ com-

pared to FrugalGPT, 33% – 92% compared to LLMSelector, and 83% – 94% compared to ELLM, while incurring comparable or only slightly higher costs compared to GraphRouter and CLLM.

Furthermore, MixLLM can simultaneously improve result accuracy and significantly reduce inference cost, it achieves 1.1% – 5.9% higher accuracy than FrugalGPT while reducing inference cost by 39% – 78%; 0.8% – 9.8% higher accuracy than LLM-Selector with a 21% – 92% cost reduction; and 1.1% – 5.9% higher accuracy than ELLM while decreasing inference cost by 68% – 86%. In summary, MixLLM demonstrates its practicality and robustness across diverse workflows and datasets for the same tasks.

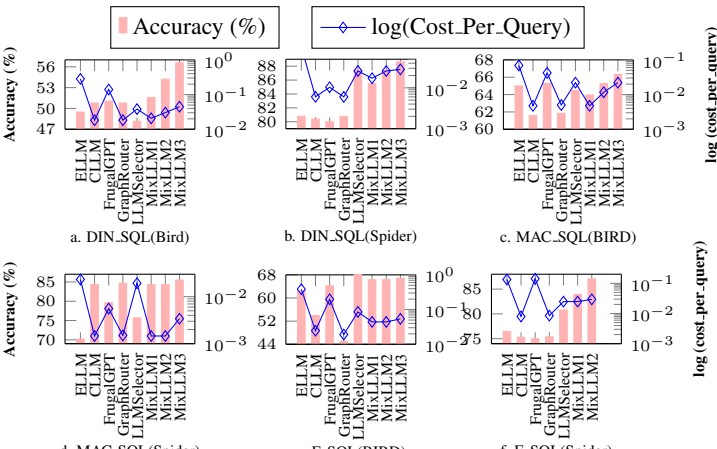

Figure 5: MixLLM achieves superior cost-accuracy trade-offs across various workflows and datasets in the text-to-SQL task.

### 4.5 ABLATION STUDY

**Effect of Two-phase Greedy Select.** We evaluate the effectiveness of the two-phase greedy selection strategy by replacing the original selection strategy in MixLLM with a UCT-based selection strategy while retaining all other details, the derived MixLLM-G approach inherits the capability of automatic early termination. Thanks to this capability, MixLLM-G and MixLLM finally identify LLM combinations with identical result quality and inference cost, but they diff from exploration efficiency. Therefore, we primarily compare their exploration efficiency.

Figure 6(a) and Figure 6(b) compare the performance of MixLLM and MixLLM-G under two complementary exploration efficiency metrics: (1) i-th exploration, which denotes the number of exploration taken to find the final solution; and (2) normalized exploration cost, which is the normalized total cost incurred during exploration. In Figure 6(a), lower bars are better, indicating an earlier appearance of the final optimal solution. MixLLM discovers optimal solutions earlier than MixLLM-G across all evaluated tasks, this attributes to two-phase greedy selection strategy, which dynamically balances accuracy and cost to facilitate the earlier discovery of the optimal solution. Figure 6(b) presents the normalized exploration cost. Compared to MixLLM-G, MixLLM significantly reduces the exploration cost by 17% to 64%. This is because the earlier discovery (*cf.* Figure 6(a)) of the final optimal solution accelerates the identification and pruning of unpromising nodes with high cost or low quality, thus reducing unnecessary exploration.

**Automatic Termination w/o Cost or Accuracy based Optimization.** Figure 7 compares MixLLM with *MixLLM-CE* and *MixLLM-AP*, which terminate without identifying unpromising nodes based on estimated cost (*i.e. MixLLM-CE*) or predicted accuracy (*i.e. MixLLM-AP*). We mainly compare exploration cost, because all of these approach finally find LLM combinations with identical result quality and inference cost due to automatic termination mechanism. Compared to MixLLM-CE and MixLLM-AP, MixLLM reduces the total exploration cost by 56% – 72% and 0% – 45%, respectively. These performance improvements benefit from the hybrid pruning strategy that takes advantage of both estimated cost and predicted accuracy to prune unpromising nodes, which is more effective than relying solely on accuracy (*i.e.* MixLLM-CE) or cost (*i.e.* MixLLM-AP).

### 4.6 ERROR ANALYSIS: COST AND ACCURACY BASED NODE IDENTIFICATION AND PRUNING

Figure 8a presents the cost estimation errors (*i.e.* $\frac{|cost_{est}-cost_{real}|}{cost_{real}}$) and the cumulative probability distributions for the explored LLM combinations across various tasks. Four tasks exhibit less than

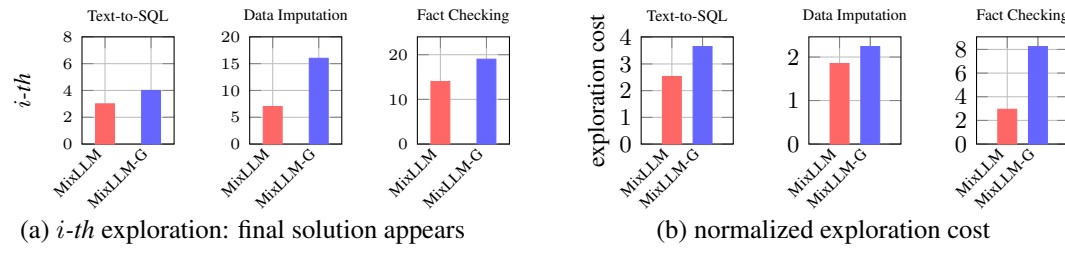

(a) *i-th* exploration: final solution appears      (b) normalized exploration cost

Figure 6: Exploration efficiency of MixLLM is improved by two-phase greedy strategy.

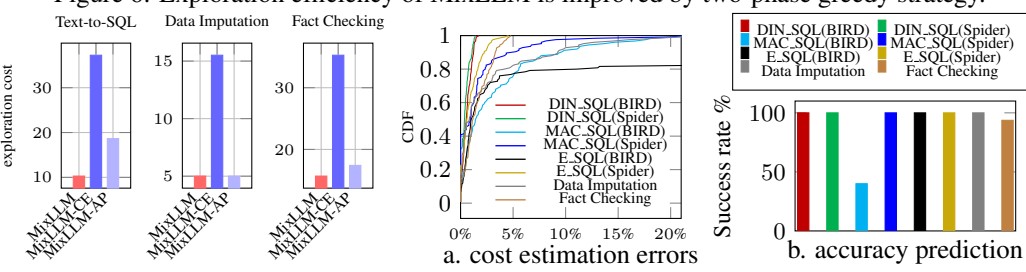

Figure 7: Ablation study of automatic termination .

Figure 8: Cost estimation error (a) and accuracy prediction success rate (b).

5% cost estimation errors for all LLM combinations, three additional tasks maintain less than 10% errors for 90% of combinations and one task has less than 20% errors in 80% of combinations. This further validates the effectiveness of identifying and pruning unpromising node via cost estimation.

Figure 8b shows the success rate of pruning unpromising LLM combinations according to predicted accuracy, where unsuccessful predictions occur if LLMs with satisfactory accuracy are incorrectly pruned (and vice versa). MixLLM achieves a prediction success rate of nearly 100% in all tasks, except MAC_SQL(Spider), which only has a success rate of 40%. This is due to the marginal accuracy difference ($<=2\%$), coupled with the limited empirical runtime data for prediction, which makes it difficult to discern such subtle variations.

## 5 CONCLUSION

In this paper, we investigate the actual performance characteristics of LLMs in the multi-stage complex tasks based on extensive experiments and have interesting findings. One of the most important findings is that employing different LLMs for multi-stage tasks can simultaneously improve result quality and reduce LLM inference cost. This greatly motivates us to perform this research work. We present all our findings in this paper, hoping to boost the development of related fields. To the best of our knowledge, we are the first to present a practical and comprehensive LLM performance summary on multi-stage tasks.

Furthermore, we address the challenge of selecting LLMs for multi-stage complex tasks, aiming to jointly optimize final result quality and inference cost. Inspired by observed performance characteristics of LLMs, we formalize the problem from a practical perspective and propose MixLLM, a non-training tree-based search approach that adaptively balances result quality and cost, and supports automatic early termination. Through experiments on multiple diverse multi-stage workflows and datasets across three different tasks, and a variety of state-of-the-art LLMs and baselines, we demonstrate the superiority of MixLLM over baselines. MixLLM can improve result accuracy by $1\% - 16\%$ and simultaneously reduce inference cost by $18\% - 92\%$.

In the future, MixLLM could be extended to efficiently handle a wide range of LLMs, as commonly found in production environments where the candidate LLMs can reach up to 60 or more. To address the resulting search overhead, integrating an LLM filtering mechanism to pre-select cost-effective candidates is a promising direction.

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

## A  FINDINGS: PERFORMANCE CHARACTERISTICS OF LLMs IN MULTI-STAGE TASKS

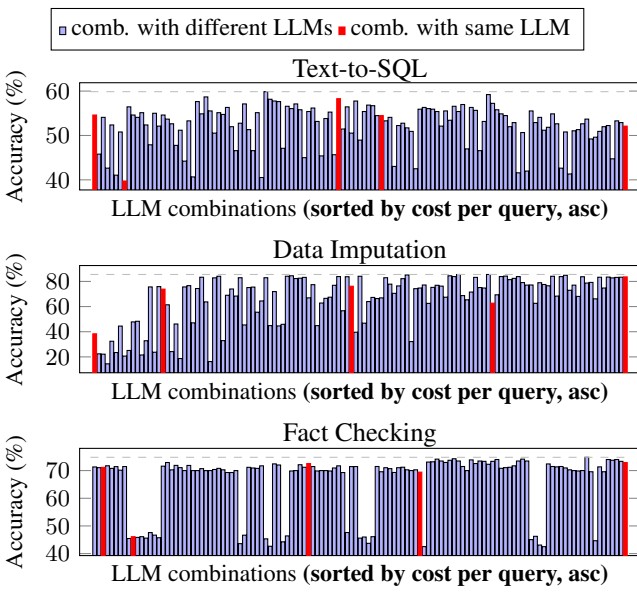

Figure 9: An extention to Figure 1(b): the best combination with the same LLM is not the clear winner.

We perform a comprehensive study to explore the performance characteristics of LLMs in multi-stage tasks, see details of tasks, workflows and datasets in Section 4. We summarize our findings as follows.

**Finding 1: It is better to apply different LLMs rather than the same LLM in terms of result quality and LLM cost.**

In Figure 9, the red and blue bars represent the combinations with the same LLM or various (possibly different) LLMs, respectively. It is clear that the combination across all LLMs with the highest

accuracy ($best_{all}$) performs better than the combination of the same LLM with the highest accuracy ($best_{same}$) in terms of accuracy and cost. Specifically, the accuracy is improved by 1.58% to 1.92% and the cost is simultaneously saved by 20.27% to 53.85%, according to the details in Table 2. In addition, for each task, among the accuracy interval between $best_{same}$ and $best_{all}$, we also observe many other combinations with different LLMs that significantly reduce the cost. The cheapest combination among which saves up to 36%, 93%, 85% of the cost compared to $best_{same}$ in the three representative tasks, respectively.

Table 2: Performance comparison: $best_{same}$ (the combination of the same LLM with the highest accuracy) vs. $best_{all}$ (the combination across all LLMs with the highest accuracy).

| Multi-stage Tasks | $best_{same}$ acc — cost | $best_{all}$ acc — cost | benefit of *hybrid LLMs* acc($\uparrow$) — cost ($\downarrow$) |
| --- | --- | --- | --- |
| Text-to-SQL | 58.29% — $0.074 | 59.87% — $0.059 | 1.58% — 20.27% |
| Data Imputation | 83.60% — $0.104 | 85.52% — $0.048 | 1.92% — 53.85% |
| Fact Checking | 72.92% — $0.040 | 74.80% — $0.027 | 1.88% — 32.5% |

**Finding 2: The performance of LLM combinations varies across different workloads.**

The accuracy of LLM combinations varies by tasks, for example, for the three tasks, the $best_{all}$ combinations are different. In addition, the cost of LLM combinations also varies by tasks. This is because: 1) the numbers of tokens generated by LLMs are different (due to the difference of characteristics in LLMs) ; and 2) LLMs have varying pricing strategies. We have similar findings on different methods and datasets (see details in our evaluation results in Section 4).

> From Finding 1 and 2, it is clear that simply applying the same fixed LLM to all stages is not a good choice. It is necessary to design an effective LLM selection methodology that could find hybrid LLMs to improve result quality and reduce cost and adapt to various workloads.

**Finding 3: There is no clear relationship between the result quality and cost of LLM combinations.**

Figure 9 also illustrates the relationship between result quality and LLM cost. In the figure, from left to right along the x-axis, the cost is increasing, but the accuracy of LLM combinations sometimes increases and sometimes decreases. The cost of $best_{all}$ is neither the highest nor the lowest. In Table 2, we also find that expensive LLM combinations might have lower accuracy and vice versa.

**Finding 4: The best combination with the same LLM can produce relatively high-quality result, but might be expensive.**

Through an in-depth comparison between $best_{same}$ and $best_{all}$ in Figure 9 and Table 2, we find that their difference in accuracy is much smaller than the difference in cost. Specifically, the accuracy of $best_{same}$ is close to that of $best_{all}$, with a loss of less than 2%, while the cost can be saved by up to more than 50%. Notably, although different LLMs exhibit performance variance on different tasks, some LLMs, including $best_{same}$, are more suitable to handle this task than others. As a result, the performance of $best_{same}$ could reach a reasonably good level, even may not optimal. Meanwhile, the cost of $best_{same}$ is much lower than that of $best_{all}$.

> From Finding 3 and 4, it is very challenging to measure the goodness of LLM combinations and then search for the optimal one. Luckily, there is an easy way to obtain a *base solution* (*i.e.*, $best_{same}$) to specify a criteria for high quality result. Taking this criteria in hand, it is more efficient to find other combinations of LLMs with comparable quality and less cost than exhaustive search with no guidelines.

Next, we examine the intrinsic properties of LLMs that may affect our problem definition.

**Finding 5: the result accuracy of LLM fluctuates within a range over multiple experiments.**
In Figure 1(c) and (d), each box represents the performance of an LLM combination over multiple

experiments under the same settings. From Figure 1(c), we see that the accuracy of an LLM combination fluctuates within a range, regardless of whether to use the same LLM or hybrid LLMs. This is caused by the internal stochasticity nature of LLMs (Vaswani et al., 2017). In addition, we study the performance of the same LLM (*i.e.*, GPT-4o) under multiple rounds of experiments. In each round, we repeat the same experiment to answer text-to-SQL queries five time. Figure 1(d) presents the result in four rounds, we see that the fluctuation ranges of different rounds overlap significantly and the mean values are close, indicating that the accuracy of LLMs fluctuates within a certain range and this variability is relatively stable. It would be better to define the LLM selection problem in a practical way that tolerates the performance fluctuation of LLMs.

# B  PSEUDO-CODE OF THE EXPAND PHASE IN MIXLLM

---
**Algorithm 2:** Expand

---
1 **Function** *Expand* *(task $T$, set of LLMs $\mathcal{L}$, selected node $v = (v_1, \ldots, v_k)$)*
2     **for** *each stage $j$ in task $T$* **do**
3        **for** *each LLM $\ell \in \mathcal{L}$ and $\ell \neq v_j$* **do**
4           $u = (v_1, \ldots, v_{j-1}, \ell, v_{j+1}, \ldots, v_k)$
5           **if** *$u$ is not a child of any node in $T_S$* **then**
6              Add $u$ as a child of $v$

---

