# OpenReview forum: "MixLLM: Selecting Large Language Models with High-Quality Results and Minimum Inference Cost for Multi-Stage Complex Tasks"
_ICLR.cc/2026/Conference — Submitted to ICLR 2026_

### Official Review · Reviewer_Wf1K · 2025-10-17

**Soundness:** 2
**Presentation:** 2
**Contribution:** 3
**Rating:** 4
**Confidence:** 4

**Summary:**

This paper addresses the problem of selecting appropriate Large Language Models for multi-stage complex tasks to jointly optimize result quality while minimizing inference cost. The authors formalize this as a constraint-based optimization problem with the intention to guarantee lower bounds on accuracy and minimize LLM inference cost. They propose MixLLM, a training-free tree-based search framework that leverages exploration-exploitation principles with a two-phase greedy strategy to adaptively balance result quality and cost. The method includes automatic early termination through search-space pruning based on cost estimation and accuracy prediction.

**Strengths:**

1. The problem formulation is technically sound and practical. The constraint-based single-objective optimization approach with the relaxation parameter $\theta$ elegantly handles LLM stochasticity, which is empirically validated through multi-round experiments showing stable fluctuation ranges.
2. The technical contributions are substantial. The two-phase greedy selection strategy that switches from accuracy prioritization to cost minimization once the accuracy constraint is satisfied is intuitive yet effective.
3. Unlike competing methods (FrugalGPT, GraphRouter, HybridLLM, RouteLLM) that require training routers on task-specific data, MixLLM is training-free and automatically terminates when no promising nodes remain. This is a significant practical advantage, reducing deployment barriers and making the approach more generalizable across different domains.

**Weaknesses:**

1. My major concern is, while the paper claims the method is efficient, there is insufficient analysis of computational overhead and scalability. The candidate LLM pool is limited to 5 models, and for a k-stage task, the combinatorial search space is |L|^k. It is not clear how the proposed approach can scale to thousands of LLMs.
2. The accuracy prediction mechanism shows concerning limitations. For MAC-SQL(Spider), the prediction success rate is only 40%, attributed to marginal accuracy differences (≤2%) and limited empirical runtime data. This suggests the prediction mechanism may fail precisely when fine-grained distinctions matter most. The paper does not provide: (a) detailed description of the prediction algorithm beyond high-level concepts about "runtime experience knowledge" and "node lineage," (b) ablation studies isolating the accuracy prediction component's contribution, (c) analysis of failure modes and when the predictor can be trusted, or (d) comparison with standard supervised learning baselines for accuracy prediction.
3. The related work section omits several relevant areas that diminish the paper's positioning. Missing categories include  (a) Mixture-of-Experts (MoE) literature for routing between specialized models, which addresses similar selection problems, (b) LLM selection [1] and routing works [2], and more.
4.  The paper lacks systematic analysis of when and why the method works. Key missing elements include: (a) characterization of task properties (e.g., stage dependencies, error propagation patterns, intermediate result quality distributions) that influence LLM selection effectiveness, (b) theoretical analysis or convergence guarantees for the search procedure.


[1] RELM: https://openreview.net/pdf?id=gWi4ZcPQRl
[2] Smoothie: https://openreview.net/pdf?id=pPSWHsgqRp

**Questions:**

Q0: Answer the concerns raised in the Weaknesses section above.
Q1: The candidate LLM pool in your experiments is limited to 5 models, yet you mention production environments may have 60+ candidate LLMs. Can you provide empirical analysis of: (a) wall-clock time for the search process across different pool sizes, (b) average number of examined nodes before termination, (c) how performance degrades with LLM pools of 10, 20, and 40+ models, and (d) what percentage of inference cost savings is consumed by exploration overhead?
Q2: Can you provide: (a) theoretical guarantees on solution quality (e.g., approximation bounds showing the found solution is within X% of optimal), (b) convergence analysis or proof that the method terminates in finite time under all conditions, (c) worst-case complexity analysis for the number of examined nodes, and (d) conditions under which the method is guaranteed to find the global optimum?
Q3: For practitioners deploying MixLLM, how should they determine: (a) the baseline accuracy $a$ when there is no clear "best same-LLM" combination or when different same-LLM combinations perform similarly, (b) the relaxation parameter $\theta$, (c) thresholds for pruning nodes in the cost estimation and accuracy prediction mechanisms.

---

### Official Review · Reviewer_3T8d · 2025-10-27

**Soundness:** 2
**Presentation:** 3
**Contribution:** 2
**Rating:** 4
**Confidence:** 3

**Summary:**

This paper addresses the challenge of optimizing both quality and inference cost when deploying large language models for multi-stage complex tasks such as text-to-SQL, data imputation, and fact checking. The authors propose MixLLM, a search-based framework that dynamically selects model combinations for each stage under a relaxed accuracy constraint. The method introduces a two-phase greedy strategy that first ensures the accuracy threshold and then minimizes cost, aided by lightweight predictors for accuracy trend and cost estimation. The framework incorporates pruning and automatic early stopping to reduce exploration overhead. Experiments on multiple datasets and workflows show that MixLLM achieves comparable or better accuracy while cutting inference costs.

**Strengths:**

- The problem formulation is well motivated and practical, focusing on the trade-off between LLM performance and inference cost in realistic multi-stage pipelines where different subtasks may favor different models.
- The overall algorithmic design is coherent: the two-phase greedy mechanism, early termination rule, and heuristic pruning collectively yield an elegant balance between exploration and efficiency.

**Weaknesses:**

- Although MixLLM adopts a “two-phase greedy” strategy combined with pruning and early stopping, the paper provides no theoretical analysis of its search efficiency or convergence behavior, nor any discussion of computational complexity such as bounds on the number of node evaluations. For a framework that claims to perform near-optimal search, the absence of formal optimality guarantees or error bounds makes the theoretical foundation relatively weak.
- The accuracy predictor operates as a binary classifier that only captures “better or worse than parent,” which may oversimplify model behavior; richer regression or uncertainty-aware predictors could yield more stable guidance for node selection.
- The assumption of pre-defined task decomposition (fixed multi-stage workflows) constrains generality—MixLLM does not address how to automatically decompose or adapt workflow structures, which would be critical for broader applicability.

**Questions:**

See #Weaknesses

---

### Official Review · Reviewer_Ybpc · 2025-10-28

**Soundness:** 2
**Presentation:** 2
**Contribution:** 3
**Rating:** 4
**Confidence:** 3

**Summary:**

The paper tried to optimize selection of LLMs for multi-stage complex tasks to jointly optimize accuracy and minimize LLM inference cost. The problem is formalized as a constraint-based optimization task. The paper introduces MixLLM, that is, a framework for finding the combination that balances result quality and inference cost. MixLLM is a tree-based search method that treats LLM performance as black box, and utilizes a two phase greedy strategy. Paper evaluates on three tasks: text-to-SQL, data imputation, and fact checking, and compares against LLM selectors and routers.

**Strengths:**

Originality: Addresses an important problem of selecting models under cost/quality trade-offs via a black-box formulation that avoids strong assumptions and can flexibly explore model pipelines.


Quality: Empirically beats baselines consistently across tasks, indicating robust performance gains.


Clarity: Ablation studies cleanly isolate the contribution of each component, making the source of improvements credible.


Significance: The assumption-free, black-box setup makes the approach practical.

**Weaknesses:**

Clarity. Several key notions are insufficiently defined or explained. In particular, the paper does not clearly specify how cost and performance are estimated, what constitutes “(near-)optimal” selection, or how end-to-end cost is computed when the search/selection overhead is included. Please see the questions below for concrete clarification requests.

Limited scalability/analysis. The evaluation considers only five models, which is small relative to realistic settings where practitioners may select from dozens to hundreds of candidates. The paper does not quantify the wall-clock overhead of the search nor analyze how this overhead scales with the number of models, creating ambiguity about when the method might become more expensive than simply running a top model outright. Reporting only cost-per-query (rather than wall-clock time) further obscures practical deployment costs.

**Questions:**

1. Model-pool size: Results use 5 models. How does performance and cost change with larger pools (e.g., 50–100), as explored in related model-selection literature [2]?


2. Overhead (wall-clock): What is the end-to-end wall-clock overhead of your method?


3. Scaling behavior: How does overhead scale with the number of models and the exploration budget? Where is the point beyond which it is cheaper to run the most capable (and costly) model directly?


4. End-to-end cost: Please report inference cost with search overhead included (not just per-query cost without selection time), so one can assess true deployment cost.


5. Cost/performance estimation: How are costs (tokens, API pricing, compute time) and performance (metrics, confidence intervals) estimated during search? Are these online estimates unbiased, and how are their variances handled?


6. “(Near-)optimal” definition: What exactly is considered (near-)optimal in your paper? Please formalize this notion.


7. Reporting metric: Why report cost-per-query instead of wall-clock time (or both)? Wall-clock is often the operational constraint in production.


8. Comparative baselines: Can you compare against OCCAM [1] (or justify its omission)?


[1] Ding, Dujian, Bicheng Xu, and Laks VS Lakshmanan. "Occam: Towards cost-efficient and accuracy-aware classification inference." arXiv preprint arXiv:2406.04508 (2024).

[2] Karimi, Mohammad Reza, et al. "Online active model selection for pre-trained classifiers." International Conference on Artificial Intelligence and Statistics. PMLR, 2021.

---

### Official Review · Reviewer_xeBm · 2025-10-31

**Soundness:** 3
**Presentation:** 2
**Contribution:** 2
**Rating:** 2
**Confidence:** 4

**Summary:**

Motivated by a series of empirical observations on LLM performance characteristics in multi-stage tasks, this paper investigates the quality–cost trade-off in selecting LLMs across different task stages and identifies four key challenges. To address these challenges, the paper first formulates the multi-stage LLM selection problem and then proposes a tree-based search framework named MixLLM. This framework employs a simple yet effective two-phase greedy strategy to balance accuracy and computational cost. Once accuracy is ensured, the strategy further minimizes cost through automatic and early termination.

**Strengths:**

1. The paper formulates a new problem of optimizing LLM routing across multi-stage tasks, aiming to select appropriate LLMs for each stage to achieve qualified high-quality final results while reducing cost as much as possible.
2. The experimental results are solid. MixLLM demonstrates superior trade-offs between accuracy and inference cost compared with several strong baselines, with consistent improvements observed across multiple tasks and datasets.

**Weaknesses:**

1. The presentation requires significant improvement. (1) The Introduction section fails to provide a clear overview of the method’s high-level design. (2) The discussion of related work is insufficient. The paper focuses solely on binary routers while overlooking other routing strategies, such as non-predictive methods (e.g., [1,2]) and predictive routing approaches (e.g., [3]). (3) The paper lacks sufficient details about the methodology and experimental setup, which limits reproducibility. For example, it is unclear how the costs of open-source LLMs are measured when executed locally, and how the multi-stage tasks are practically constructed. (4) The problem formulation and challenges are too coarse and insufficiently justified. For instance, the motivation for introducing Challenge C1 is unclear. Regarding C2, the authors claim that inter-stage relationships affect the optimization direction, yet the paper neither defines the optimization objective beforehand nor provides supporting evidence. Overall, the challenges are described at a superficial level; finer-grained analyses are needed, for example, examining how earlier subtasks influence the quality–cost trade-off in later stages and verifying this empirically on simple two-stage tasks. Furthermore, C3 should not attribute modeling difficulty solely to LLM complexity, as subtask complexity is also a contributing factor. Finally, C2 and C4 appear more as motivations for the problem rather than genuine methodological challenges.

[1] Tryage: Realtime, intelligent routing of user prompts to large language models
[2]   Routing to the expert: Efficient reward-guided ensemble of large language models.
[3]   Tabi: An efficient multi-level inference system for large language models

2. The problem setting assuming independent cost computation across stages is concerning. In practice, the token cache from the previous stage can be reused to reduce the inference cost of the next stage when the same model is used. However, the paper treats the cost identically regardless of whether adjacent stages employ the same or different LLMs.

3. The paper does not clearly introduce the unique challenges arising from the proposed problem. The authors extend single-stage tasks to multi-stage settings and generalize LLMSelector from accuracy optimization to a quality–cost trade-off. However, the distinct challenges specific to this new formulation are not sufficiently emphasized. As a result, the proposed method appears to be a straightforward adaptation of existing designs, such as Tree-of-Thoughts, rather than a principled solution derived from the identified problem characteristics.

4. The problem formulation appears to overlook the interdependence among subtasks. It is unclear why the overall task quality varies under different cost settings. The root cause likely lies in the subtasks: the same-cost LLMs can exhibit varying performance across different subtasks, and different LLMs possess different abilities to handle the high- or low-quality outputs produced by preceding stages.

5. The name MixLLM has already been used in [4]; therefore, the paper should adopt a different name to clearly distinguish its method from existing work.

[4] Wang, Xinyuan, et al. "MixLLM: Dynamic Routing in Mixed Large Language Models." Proceedings of the 2025 Conference of the Nations of the Americas Chapter of the Association for Computational Linguistics: Human Language Technologies (Volume 1: Long Papers). 2025.

**Questions:**

1. MixLLM includes co-optimization of accuracy and costs. Why does the contribution part define it as a single-objective optimization problem?
2. How are the inference costs of open-source LLMs measured when running locally? Are they converted into monetary cost or only based on token usage?
3. How were the multi-stage tasks constructed? More details are expected to be demonstrated in Appendix.
4. Qwen-2.5 is a model series rather than a single model. In experiment, is all Qwen-2.5 series used or is only a specific model used?
5. What are the criteria for choosing the candidate LLM? Could the authors consider exploring more diverse combinations, such as models from the same series with different sizes, or models with different cost? It would be interesting to see whether different model combinations affect the overall performance of MixLLM.
6. What is the hyperparameter α set in MixLLM? Does changing α affect the trade-off between accuracy and inference cost?

---

### Meta-Review · Area_Chair_Ld9p · 2026-01-03

**Summary:**

This paper studies LLM selection for multi-stage tasks under a quality-cost trade-off and proposes the MixLLM framework. However, the authors did not participate in the rebuttal, and none of the reviewers’ major concerns were addressed. These outstanding issues include insufficient methodological clarity, weak problem formulation and justification of challenges, lack of scalability and overhead analysis, unclear cost modeling, limited theoretical grounding, and positioning relative to prior routing and selection work. Given the absence of author responses and the persistence of substantial reviewer concerns, I recommend reject.

**Reviewer Concerns:**

The authors did not provide a rebuttal. As a result, none of the substantive reviewer concerns were addressed.

**Reviewer Scores:**

The authors did not provide a rebuttal. As a result, none of the scores may be changed.

---

### Decision · Program_Chairs · 2026-01-26

Reject